# Current Advances in Humanized Mouse Models for Studying NK Cells and HIV Infection

**DOI:** 10.3390/microorganisms11081984

**Published:** 2023-08-02

**Authors:** Jocelyn T. Kim, Gabrielle Bresson-Tan, Jerome A. Zack

**Affiliations:** 1Department of Medicine, Division of Infectious Diseases, University of California Los Angeles, Los Angeles, CA 90095, USA; jocelynkim@mednet.ucla.edu (J.T.K.);; 2Department of Microbiology, Immunology and Molecular Genetics, University of California Los Angeles, Los Angeles, CA 90095, USA; jzack@ucla.edu; 3Department of Medicine, Division of Hematology and Oncology, University of California Los Angeles, Los Angeles, CA 90095, USA

**Keywords:** humanized mice, BLT, HIV, AIDS, NK cells

## Abstract

Human immunodeficiency virus (HIV) has infected millions of people worldwide and continues to be a major global health problem. Scientists required a small animal model to study HIV pathogenesis and immune responses. To this end, humanized mice were created by transplanting human cells and/or tissues into immunodeficient mice to reconstitute a human immune system. Thus, humanized mice have become a critical animal model for HIV researchers, but with some limitations. Current conventional humanized mice are prone to death by graft versus host disease induced by the mouse signal regulatory protein α and CD47 signaling pathway. In addition, commonly used humanized mice generate low levels of human cytokines required for robust myeloid and natural killer cell development and function. Here, we describe recent advances in humanization procedures and transgenic and knock-in immunodeficient mice to address these limitations.

## 1. Introduction

There are approximately 38 million people currently living with HIV/AIDS worldwide. Anti-retroviral therapy (ART) suppresses HIV replication in people living with HIV/AIDS for years such that viremia is largely undetectable using standard clinical assays [1,2,3]. However, ART is not curative due to the latent viral reservoir. HIV infects and stably integrates a full-length copy of the replication-competent viral genome into the host chromosome of target cells such as CD4+ T cells and macrophages [4]. The most well-defined and prominent ‘latent reservoir’ is within memory CD4+ T cells [5,6]. During HIV infection, a subset of these infected CD4+ T cells avert virus-induced death and transition into memory cells [5,6]. Because memory cells are by nature long-lived and quiescent, these infected cells can persist for decades with their silenced proviral cargo, only to later produce an infectious virus if ART is discontinued [7,8,9]. Due to their longevity, latently infected cells decay very slowly during ART [5]. In addition, recent studies profiling HIV-1 integration sites demonstrated that infected cell clones with identical integration sites could persist for more than 11 years and proliferate over time [10,11]. These breakthrough studies demonstrated that despite effective ART, the latent reservoir can proliferate and increase over time.

In vitro experiments involving cell lines, primary cells, and primary tissues have been critical to understanding HIV biology. Ex vivo organoid models have emerged as a valuable alternative to study HIV neuropathology [12] due to the limited availability of human primary neuronal cells and post-mortem tissue. However, it can be difficult to study HIV pathogenesis and complex immune interactions with only in vitro or ex vivo systems. The most common in vivo models for HIV research utilize humanized mice or non-human primates (NHP). Because NHP are physiologically and immunologically similar to humans, the simian immunodeficient virus (SIV) NHP model [13] has been important for the preclinical evaluation of drugs, vaccines and gene-therapies against HIV, and the study of lentiviral immunity in vivo. However, NHP studies can be costly, lengthy, and vulnerable to monkey shortages [14], making their common use prohibitive. An alternative in vivo model involves the transplantation or introduction of various human immune tissues or cells into immunodeficient mice [15]. Due to the human species tropism of HIV, humanized mice are a widely used small animal model for HIV research. Over the past several decades, different immune-deficient mouse strains have been developed and tested for human immune engraftment and are reviewed elsewhere [15,16,17]. R5 and X4 tropic HIV isolates replicated efficiently in humanized mice (as reviewed previously [18]) and were suppressed by various regimens of ART administered via intraperitoneal injections [19,20,21,22], subcutaneous injection [23,24,25], drinking water [21,26], or animal feed [23,24,27,28,29]. Latent HIV reservoirs were found in various anatomic sites in the infected humanized mice [20,21,26,27,30], with evidence of viral rebound following ART interruption, similar to the rebound viremia seen in people living with HIV if ART is discontinued. This review will outline recent advances in humanized mouse models in the field of HIV research with a particular focus on improving natural killer (NK) cell reconstitution.

## 2. Commonly Used Humanized Mouse Models

### 2.1. NSG, NRG, NOG, BRG Mice

The most common approach to creating a humanized mouse model is transferring human immune cells or tissue into immunodeficient mice. This allows for the engraftment of functional human immune cells in the murine context to study various human infections. The most widely used immunodeficient mice are NOD.Cg-*Prkdc^scid^ Il2rg^tm1Wjl^*/SzJ (NSG), NOD.Cg-*Rag1^tm1Mom^ Il2rg^tm1Wjl^/SzJ* (NRG), NOD.Cg-*Prkdc^scid^ Il2rg^tm1Sug^*/JicTac (NOG), and C;129S4-*Rag2^tm1.1Flv^ Il2rg^tm1.1Flv^*/J (BRG) as listed in Table 1 and Appendix A. These mice harbor mutations in either the recombination activating gene (Rag) or protein kinase catalytic polypeptide (*Prkdc^scid^*) resulting in the loss of endogenous mouse T cells and B cells. The additional mutation in the *IL2RG*, which encodes the interleukin-2 receptor subunit γ (IL2rγ), results in deficiencies in multiple cytokines signaling pathways such as IL-2, IL-4, IL-7, IL-9, and IL-15. The loss of murine IL-15 signaling leads to poor murine NK cell development. Thus, the disruption in IL2rγ signaling allows for an increase in tolerance to xenotransplantation. In addition, these mice are on a NOD background, which have inherent deficiencies in mouse macrophage, dendritic cell, NK cell, and the complementary system. The most common approaches to humanization of mice are detailed in this review.

### 2.2. HuPBL and PDX-NSG Mice

The first model of humanized mice was generated via injection of human peripheral blood lymphocytes (huPBL) into immunodeficient SCID (C.B-17*scid*) mice over 35 years ago [33]. Human lymphocytes quickly engrafted as soon as two hours after PBL injection and demonstrated an activated phenotype [33]. Given that huPBL-SCID mice were rapidly engrafted with mature human CD4+ T cells, HIV could be administered two hours after PBL injection resulting in robust viral replication and CD4+ T cell decline in vivo. In a later variation of this model, human peripheral lymphocytes from adult donors were injected intraperitoneally into immunodeficient NSG mice (huPBL-NSG), which had an additional *IL2RG* mutation, leading to improved human immune engraftment [31,32], likely due to the lack of murine NK cells that could inhibit xenogeneic cell engraftment. Human lymphocytes expanded in the huPBL-NSG mouse efficiently and permitted HIV replication in vivo for extended periods of time. However, human myeloid and NK cells lacked proliferative capacity resulting in low reconstitution and contributed to minimal to absent primary immune responses.

Recently, human lymphocytes containing latently infected cells from long-term ART-suppressed patients were used to humanize NSG mice to generate patient-derived xenograft (PDX) mice [52]. These mice developed viremia 2 weeks after PBL injection, indicating the rapid reactivation of proviruses and an in vivo model to study the viral reservoir. However, these humanized mice quickly succumbed to graft versus host disease (GvHD). The rapid development of GvHD in huPBL mice is caused by the xenogeneic reaction of human cells toward mouse tissues, resulting in severe morbidity and mortality, thus limiting their use for long-term in vivo studies.

Interestingly, studies in human patients undergoing allogeneic stem cell transplantation noted that the engraftment of human naïve CD4+ T cells, but not CD4+ memory T cells, were triggering GvHD [53,54,55]. To this end, CD4+ memory T cells from ART suppressed patients were transferred into NSG mice to create patient-derived xenograft (PDX) mice. Importantly, these PDX mice could be followed for approximately eleven weeks without developing GvHD [35]. In addition, approximately eight weeks after naïve CD4+ T cell injection, viremia was detected indicating the reactivation of proviruses from a viral reservoir. In addition, graft-autologous HIV antigen-specific CD8 T cells clones demonstrated viral control, followed by the subsequent generation of viral escape mutations. Thus, the PDX mice transplanted with human naïve CD4+ T cells from ART suppressed patients offers a way for future studies to investigate the viral reservoir after ART interruption and antigen-specific immune responses.

### 2.3. HuCD34 Mice

Another model of humanized mice is HuCD34 mice, which consists of human CD34+ hematopoietic stem cells (HSCs) injected into immunodeficient mice (e.g., NSG, NRG, NOG, and BRG). These CD34+ HSCs can be processed from human fetal liver or non-fetal human sources such as umbilical cord blood (UCB), adult bone marrow, or granulocyte colony stimulating factor (G-CSF) mobilized peripheral blood stem cells. Varying levels of lymphoid, myeloid, and erythroid lineages may develop depending on the cell source utilized [36]. CD34+ HSCs are most commonly injected intravenously into adult mice or intra-hepatically into newborn mice [37,38]. Prior to transplantation, immunodeficient mice are often conditioned with myeloablative sub-lethal irradiation or chemotherapy (e.g., busulfan) prior to transplantation. Transplanted human CD34+ HSCs have differentiated into functional human T and B cells allowing for efficient HIV replication in vivo [38,39,40,41] via rectal or vaginal mucosal challenges [56]. In addition, HuCD34-NOG mice were capable of anti-HIV IgG responses, but with limited breadth to HIV antigens [42]. Additional advances have improved antigen-specific IgG responses by matching human leukocyte antigen (HLA) class I and class II molecules between human HSCs and human HLA-matched transgenic mice (reviewed in [57]).

### 2.4. NSG-BLT Mice

A third method for humanization involves the surgical implantation of matching human fetal thymus and liver fragments as well as injection of matched CD34+ HSCs into conditioned immunodeficient adult mice to create bone–liver–thymus (BLT) mice [30,43,44,58,59,60]. In some BLT mice, matching human fetal thymus and fetal liver-derived CD34+ HSCs are used. The BLT-NSG mouse is one of the most commonly used mouse models in HIV research due to the efficient human multi-lineage immune engraftment. The fetal human HSCs engraft and populate the mouse bone marrow, serving as progenitor cells for human lymphoid and myeloid cells. Multi-lineage reconstitution of human T cells, B cells, and myeloid cells was found throughout the mice including the human thymus and murine liver, bone marrow, thymus, spleen, lymph nodes, lungs, female reproductive tract, and gut [30,43,44,60,61,62]. NSG-BLT mice were susceptible to HIV infection via rectal and vaginal HIV challenges [46,47,62,63,64] despite lacking immune aggregates such as gut associated lymphoid tissue [65]. Because the human T cells were educated upon the background of the human thymus tissue, BLT mice demonstrated increased T cell engraftment and anti-HIV specific CD8+ T cell responses [43,45,59]. The BLT mice also demonstrated B cells, albeit mostly immature and minimal memory B cells with limited class-switching [66,67]. HIV-specific human antibodies, particularly human IgM, have been elicited after HIV infection [43,68] or immunization [69,70,71]. However, achieving robust antigen-specific humoral vaccination in BLT mice that recapitulate healthy adult human immunization responses has been difficult. Engraftment of human NK cells has been low [72], but they demonstrate in vivo function. Recently, tissue-specific functional memory NK cells were isolated from the liver of BLT-immunized mice [73]. Despite their limitations, NSG-BLT mice have been used extensively to test anti-HIV drugs, antibodies, and cellular therapies (reviewed in [15,17,74]). 

### 2.5. HuCD34-SRG, HuPBL-SRG, HuCD34-TKO, and TKO-BLT Mice

Humanized mice models utilizing the NSG, NRG, NOG, and BRG mouse strains can be restricted in their experimental timelines due to the development of GvHD [75]. Although there is biological variation between human donors and mouse series, the mice will begin to succumb to GvHD by ~20 weeks transplantation of human CD34+ HSCs or tissues. Signal regulatory protein α (SIRPα) is an immunoglobulin superfamily receptor expressed on myeloid cells and is activated by allogeneic or xenogeneic interactions with non-self CD47 [76,77]. Human CD47 has a low affinity for mouse SIRPα, which results in the phagocytosis of transplanted human cells by mouse macrophages and the induction of GvHD. Interestingly, SIRPα is polymorphic between different strains of mice, and the mouse SIRPα in B6.129S and C57BL/6 mice does not cross-react sufficiently with human CD47. To this end, efficient human immune cell reconstitution and a dramatic reduction in GvHD was achieved in humanized SRG and TKO mice, which altered the CD47/SIRPα anti-phagocytic signaling pathway (reviewed in [78]). By knocking in human SIRPA to generate *SIRPA*^h/m^
*Rag2^−/−^ Il2rg^−/−^* (SRG) mice, huCD34-SRG mice achieved robust humanization and lived for 9 months without succumbing to GvHD [79]. In addition, *CD47* was knocked out of mice deficient in *Rag1* and *IL-2Rγ* to generate B6.129S-*Rag2^tm1Fwa^CD47^tm1Fpl^Il2r^gtm1Wjl/J^* TKO mice. Transplantation of human PBLs led to efficient T cell engraftment and HIV infection in vivo [50]. Importantly, by knocking out *CD47*, these huPBL mice demonstrated a 24-day delay to the onset of mild GvHD compared to huPBL-NSG mice. In addition, human fetal tissue and autologous CD34+ HSCs were transplanted into TKO mice deficient in *Rag2, IL2rg,* and *CD47* (C57BL/6 *Rag2^−/−^γc^−/−^CD47^−/−^*) to create TKO-BLT mice. Due to the knockout of mouse *CD47* in these TKO mice, excellent and stable human immune cell engraftment was observed with little to no GVHD for up to 45 weeks after transplantation [23,51,72,80]. Indeed, these TKO-BLT mice demonstrated HIV infection and efficient suppression of viremia with ART over extended periods, which was followed by rebound viremia after ART interruption, indicating the establishment of a latent HIV reservoir in vivo. Thus, TKO-BLT mice may be placed on ART for prolonged periods, allowing for reservoir studies, which often require extended timelines [72].

### 2.6. NeoThy-NSG or NSG(W) Mice

Due to recent increased restrictions on abortion and previous changes on policies in the U.S. related to human fetal tissue research [81], the ability to efficiently source human fetal tissue to generate BLT mice may be affected at some research institutions in the future. Human non-fetal UCB CD34+ HSCs continues to be available to generate humanized mice. However, the disadvantage of using UCB is the variable and low number of CD34+ HSCs collected from a single donor unit. Thus, neonatal human thymic tissue was utilized to humanize NSG or NOD,B6.SCID *Il2rγ^−/−^Kit^W41/W41^* (NSG-W) mice with autologous or allogeneic UCB-derived CD34+ HSCs to generate NeoThy mice [48]. NeoThy mice had equivalent levels of human immune subsets including T cells compared to mice humanized with human fetal thymus and allogeneic UCB CD34+ HSCs. Further investigations are warranted to assess if these mice are efficiently infected with HIV and can establish a stable latent reservoir.

## 3. Improved Mouse Models for HIV-1 and NK Cell Research

The common and recently used mouse strains in Table 1 do not generate human cytokines and other factors such as chemokines. Although murine cells produce certain mouse cytokines such as IL-7 [82] or IL-12 [83], which can signal to human cytokine receptors, the levels of these murine cytokine levels are minimal or absent in mice with mutations in *Rag1*, *Prkdc^scid^*, and *IL2rγ* such as NSG mice. Other murine cytokines such as IL-2 [84], IL-3 [85], IL-4 [84], IL-15 [86], granulocyte–macrophage colony-stimulating factor GM-CSF [87], and macrophage colony-stimulating factor (M-CSF) [85] have little to no activity on human cells. Although transplanted human HSCs differentiate into lymphoid cells in humanized mice, reconstitution of innate immune cells is needed to help generate human cytokines and develop a more mature immune environment. Insufficient cross-species reactivity and biological function between mouse factors and their corresponding human receptors and minimal human innate immune engraftment are major factors limiting the development of a fully functional human immune system in humanized mice.

NK cells are innate immune effectors capable of intrinsically recognizing and clearing virally infected cells through multiple mechanisms. Epidemiological and genetic studies have shown NK cell interactions with self-HLA molecules are involved in recognition of HIV-infected cells and may slow disease progression, reduce viral setpoint, or mediate immune pressure [88,89,90,91,92,93,94,95,96]. IL-15 is a critical cytokine required for NK cell and T cell development, maintenance, and function. Myeloid cells can also contribute to NK cell development by providing a richer cytokine environment. However, human NK cell development and function is poor in the humanized mouse models described so far in Table 1. Due to the low level of human cytokine production, particularly IL-15, lack of cross-species IL-15 reactivity, and insufficient myeloid cell engraftment, the frequency of NK cells in HuCD34-NSG [97], NSG-BLT [73], and TKO-BLT [72,73] mice were substantially low (~1% or less of human CD45+ in the blood) compared to the expected frequency of NK cells in human peripheral blood from healthy adult donors (~5–20% of PBMCs). Although there were overall low levels of human NK cells in NSG-BLT mice, liver-specific NK cells, but not spleen or bone marrow-derived NK cells, were capable of mediating vaccination-dependent and antigen-specific killing ex vivo [73]. In addition, an injection of an IL-15 superagonist [98,99] enhanced the human NK cells present in huPBL-mice, thereby inhibiting HIV replication in vivo [100] (Figure 1). Interestingly, administration of the IL-15 superagonist also transiently reduced viral replication in SIV-infected macaques [101].

The low human NK cell engraftment in HuCD34 and BLT mice makes them an ideal model to study the effect of adoptively transferred exogenous NK cells. Although NSG-BLT mice supported transient engraftment of exogenously administered human peripheral NK cells, these NK cells inhibited viral replication in vivo [72]. In addition, a kick and kill combination, comprising a PKC modulator as a latency reversing agent (LRA) and allogenic transfusions of human peripheral NK cells from healthy donors as the killing agent, was tested in TKO-BLT mice, which were suitable for in vivo reservoir studies due to their increased longevity. The LRA and NK cell combination was administered during ART suppression in HIV-infected TKO-BLT mice and delayed viral rebound after ART interruption, reduced number of rebounding viral clones, and eliminated the reservoir in a subset of infected animals [72]. This suggests that the addition of NK cells as a “kick” step in kick and kill approaches to eliminate the HIV reservoir may be efficacious (Figure 1). However, one caveat to this approach is that a subset of NK cells can express CD4, which could make them susceptible to HIV infection [102,103,104].

To increase human NK cell reconstitution in humanized mice, multiple approaches have been explored. Direct injections of recombinant human IL-15 cytokine complexed with the IL-15 receptor α led to efficient human NK cell engraftment in humanized BALB/c *Rag2^−/−^γc^−/−^* mice with CD34+ HSCs [105]. HuCD34-BRG mice administered with exogenous human Flt3 ligand were shown to increase human myeloid and NK cell engraftment and function, suggesting an improvement in human myeloid engraftment may provide a richer human cytokine environment for human NK cell proliferation [106]. However, these exogenous human cytokine injections provided transient levels requiring repeated injections and supra-physiologic, systemic concentrations. Hydrodynamic injection of plasmid DNA encoding human IL-15 [107] or cytokine superagonist [100] was simplified to a single administration, but still led to physiologically high cytokine levels or NK cell stimulation. Other modes of long-term cytokine vector-mediated cytokine expression have been regulated by a strong human cytomegalovirus (CMV) promoter and enhancer sequence [108,109], which was associated with supra-physiologic levels of cytokines. Finally, transgenic or knock-in mice were developed, in which promoter and regulatory sequences were inserted into the mouse genome to induce more physiological expression of human cytokines. An improvement in human NK cell engraftment in humanized mice was achieved in the following transgenic and knock-in strains by expressing human IL-15 and/or improving engraftment of human myeloid cells, which express and trans-present IL-15 thereby promoting human NK cell reconstitution (Table 2 and Appendix A). Importantly, these recent humanized transgenic mice developed human NK cells that inhibited HIV-1 replication [84] and utilized antibody-dependent cellular toxicity (ADCC) via a broadly neutralizing antibody PGT121 to suppress infection [97] or CD4-induced antibodies and CD4 mimetic to control viral rebound from the reservoir [97,110] (Figure 1). Altogether, these studies demonstrate the importance of enhancing NK cell function in humanized mice.

### 3.1. NSG-Tg (IL-15) Mice

NOD.Cg-Prkdc^scid^ Il2rg^tm1Wjl^ Tg (IL15)1Sz/SzJ (NSG-Tg (IL-15)) mice expressed human IL-15 via the BAC transgenic system, in which the entire human locus including the promoter and flanking regulatory sequences were inserted into the mouse genome leading to more physiological expression. NSG-Tg (IL-15) mice were found to produce physiological serum levels of human IL-15 [111]. The injection of human PBL in these mice resulted in persistent human T, B and NK cell engraftment [112]. In addition, NSG-Tg (IL-15) mice humanized with human CD34+ HSCs supported robust and long-term human NK cell development, survival, and reconstitution in the blood, liver, spleen, bone marrow, and GI tract [119]. Additionally, splenic NK cells from HuCD34-NSG-Tg (IL-15) mice displayed significant and robust cytotoxic ex vivo function against the target K562 cell line compared to splenic NK cells from HuCD34-NSG mice. NK cells from HIV-infected HuCD34-NSG-Tg (IL-15) mice were functional with increased expression of IFN-γ, TNF-α, perforin, Ki67, and CD107a (Lamp-1)) compared to uninfected animals. The HuCD34-NSG-Tg (IL-15) mice did not develop GvHD for 6–9 months post-transplant, thus may be a suitable long-term model to study the effects of NK cells on the HIV reservoir.

### 3.2. NOG-EXL and NSG-SGM3 Mice

Recently developed transgenic mice have constitutively high levels of human IL-3 and GM-CSF to increase human myeloid and NK cell development. The transgenic NOG-EXL mice (NOD.Cg-Prkdc^scid^ Il2rg^tm1Sug^ Tg (SV40/HTLV-IL3,CSF2)10-7Jic/JicTac) ubiquitously overexpressed human *IL3* and *CSF2* under the control of the SRα promoter [114]. HuCD34-NOG-EXL mice demonstrated robust engraftment of human CD45+, T, and B cells, and particularly myeloid cells compared to control humanized non-transgenic mice [113,114]. Human NK cell engraftment was modest and <1% of CD45+ cells in HuCD34-NOG-EXL mice [114]. Acute HIV infection led to a reduction of CD4^+^ T-cells, inversion of the CD4:CD8 ratio, and a reduction of peripheral CD56^bright^ NK cells, which was similar to patterns found during natural human infection [113].

NSG-SGM3 (NOD.Cg-Prkdc^scid^ Il2rg^tmWjl^ Tg (CMV-IL3,CSF2,KITLG)1Eav/MloySzJ) transgenic mice overexpress human *IL3*, *CSF2,* and *SCF* in mice resulting in high levels of IL-3, GM-CSF, and SCF (stem cell factor), respectively [115]. HuCD34-NSG-SGM3 mice had improved myeloid, B, and T regulatory cells compared to HuCD34-NSG mice [115]. BLT-NSG-SGM3 mice showed mature B cells with antigen-specific IgG to dengue virus and Toxoplasma gondii [116,117]. Due to the improved engraftment of myeloid cells, human NK cells engraftment was also improved [117]. In addition, a HIV PDX model was generated by transplanting PBLs from ART-suppressed patients in NSG-SGM3 mice [115]. These mice developed rapid HIV reactivation off ART and significantly more activated circulating T cells compared to NSG-SMG3 mice transplanted with PBLs from uninfected donors. Due to the robust myeloid reconstitution driven by high levels of human IL3, GM-CSF, and SCF, HuCD34-SGM3 mice rapidly succumbed to GvHD-related macrophage activation syndrome, which was characterized by progressive pancytopenia, splenomegaly, and hematophagocytosis [120]. Recently, transgenic overexpression of IL-15 under the human CMV promoter was added to this strain to generate NSG-SGM3-IL15 mice. Further studies to assess NK cells and HIV infection have yet to be investigated in humanized NSG-SGM3-IL15 mice.

### 3.3. MISTRG Mice

MISTRG mice were constructed by the knock-in expression of human M-CSF, IL-3/GM-CSF, and thrombopoietin (TPO) and BAC-mediated transgenic expression of human SIRPα in Rag2^−/−^ γc^−/−^ mice [118]. HuCD34-MISTRG mice demonstrated improved human CD45+ and myeloid cell engraftment [118]. The human monocytes and macrophages were fully functional, and thus capable of trans-presenting IL-15 for the development of functional human NK cells. HuCD34-MISTRG mice also demonstrated robust T cell engraftment and were efficiently infected by X4 and R5 tropic HIV isolates [80]. However, the humanized MISTRG mice were prone to developing GvHD-related severe anemia after 20 weeks of engraftment likely due to robust macrophage development and subsequent destruction of mouse red blood cells [118].

### 3.4. SRG-15 Mice

Balb/c x 129 Rag2^−/−^ Il2rg^−/−^ hSIRPA KI hIL15 KI (SRG-15) mice were developed by knock-in replacement of mouse *IL15* by its human counterpart in SRG mice [49]. SRG-15 mice humanized with CD34+ HSCs (HuCD34-SRG-15) demonstrated similar levels of human CD45+ cells compared to HuCD34-NSG. However, due to its ability to produce human IL-15, the HuCD34-SRG-15 demonstrated significantly higher levels of peripheral blood and splenic NKp46+ NK cells compared to HuCD34-NSG, HuCD34-SRG, and HuCD34-MISTRG mice [49]. The phenotype and functional profile of NK cells from HuCD34-SRG-15 mice were similar to the human NK cell repertoire by mass cytometry. In addition, the human NK cells were capable of mediating ADCC in vivo using anti-CD20 monoclonal antibody against a xenograft B-cell tumor challenge in HuCD34-SRG-15 mice [49]. HuCD34-SRG-15 were susceptible to HIV infection and established a latent reservoir in vivo [110]. HIV-infected HuCD34-SRG-15 treated with a combination of CD4-induced antibodies and a CD4-mimetic compound, which stabilized the HIV envelope in a conformation conducive to NK cell targeting by ADCC, demonstrated diminished HIV replication, viral reservoir, and viral rebound in vivo [110]. Importantly, these studies suggest that NK cell ADCC function in SRG-15 mice can be harnessed to control and reduce HIV infection and the viral reservoir in vivo.

### 3.5. MISTRG-6-15 Mice

MISTRG-6-15 mice were created by additionally knocking in human IL-6 and IL-15 into MISTRG mice [97]. HuCD34-MISTRG-6-15 mice demonstrated significantly improved human NK cell repopulation and function in vivo compared with huCD34-NSG mice. Due to human IL-6 expression, myeloid cells engraftment also improved, which likely provided a richer cytokine environment for NK cell development and function. MISTRG-6-15 were not directly compared to MISTRG mice in this study. Human NK cells were longitudinally profiled in HIV-infected HuCD34-MISTRG-6-15 and demonstrated increased activation, proliferation, and functionality during acute infection, followed by immune exhaustion and dysfunction during chronic infection. NK cell engraftment and functionality was only partially restored by ART. These patterns of NK biology were similar to previous clinical reports delineating NK cell function from people living with HIV/AIDS [121,122,123]. Importantly, NK cell depletion by NKp46 antibody resulted in increased HIV-1 RNA levels, indicating NK cells controlled HIV-1 replication in vivo [97]. The effect these NK cells may have on the establishment and dynamics of the HIV reservoir remains to be seen.

## 4. Other Mouse Strains with Improved NK Reconstitution

The following additional recent mouse strains have been developed to enhance NK cell engraftment, but they have not been studied in the context of HIV infection (Table 3 and Appendix A).

### 4.1. NOG-IL15 Mice

BLT (NOG-IL15) mice were generated by microinjecting the human IL-15 cDNA into the mouse zygote, leading to overexpression of human *IL15* under the control of a CMV promoter [119]. The supraphysiologic levels of human IL-15 in these mice allowed for the efficient engraftment and in vivo anti-tumor function of adoptively transferred human peripheral NK cells, but ultimately led to NK cell exhaustion and dysfunction [119]. Transplantation with human CD34 HSCs and HIV infection studies have not been pursued with these mice.

### 4.2. HIL-7xhIL-15 KI Mice 

The hIL-7xhIL-15 KI mice were generated by knocking in human IL-7 and IL-15 into mouse counterpart genes of NSG mice [124]. hIL-7xhIL-15 KI NSG mice transplanted with CD34+ HSCs demonstrated abnormally high frequencies of human NK cells in the blood (43.1 ± 5.4%), spleen (28.3 ± 5.3%), and bone marrow (17.7 ± 4.7%) among the gated human CD45+ cells. The high levels of NK cell engraftment were likely due to the supraphysiological levels of human IL-15, which was regulated by a CMV promoter. To note, humanized IL-7 single knock-in mice did not demonstrate elevated frequencies of NK cells. Further studies to assess HIV infection have not been pursued with these mice.

## 5. Concluding Remarks

There are now a wide range of transgenic mice, human cell, or tissue sources, and humanization procedures to generate humanized mice. Depending on the experimental question, each model continues to have its utility. There have been recent exciting advances in the development of transgenic and knock-in mice with robust human myeloid and/or NK cell reconstitution and function. These newer humanized mice should allow researchers to better analyze the latent reservoir and assess HIV vaccines and curative therapies.

## Figures and Tables

**Figure 1 microorganisms-11-01984-f001:**
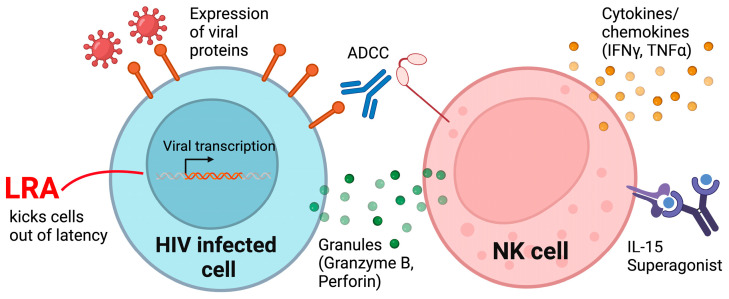
Harnessing NK cells to fight HIV infection.

**Table 1 microorganisms-11-01984-t001:** Commonly used humanized mouse strains and models.

Mouse Name	Mouse Model	Xenograft	Summary of Key Findings
NSG NRG NOG BRG	HuPBL [16,31,32,33,34] HuPDX [16,35] HuCD34 [36,37,38,39,40,41,42] BLT [30,43,44,45,46,47] NeoThy [48]	Human PBL (HuPBL, HuPDX), CD34+ HSC (HuCD34), or human thymus (NeoThy) ± human liver and CD34+ HSCs (BLT)	Minimal to low NK cell engraftment in these humanized mouse models, but appropriate model to test exogenous NK cell adoptive therapies. Prone to GvHD.
SRG	HuCD34 [49]	Human CD34+ HSC	Minimal to low NK cell engraftment. Less susceptible to GvHD due to human *SIRPA* knock-in.
TKO	HuPBL [50] BLT [23,51]	Human PBL or human thymus, liver and CD34+ HSCs (BLT)	Minimal NK cell engraftment in humanized TKO mice, but appropriate model to test exogenous NK cell adoptive therapies. Less susceptible to GvHD due to knockout of *CD47*. Mice remain healthy for 45 weeks post-humanization. Mice can be placed on ART for prolonged periods.

**Table 2 microorganisms-11-01984-t002:** Recently advanced humanized mouse models to study NK cells and HIV infection.

Mouse Name	Human Cytokines Expressed	Mouse Model	Xenografts	Summary of Key Findings
NSG-Tg(IL-15)	IL-15	HuCD34 [111,112]	Human PBL (HuPBL, HuPDX), CD34+ HSC (HuCD34), or human thymus (NeoThy) ± human liver and CD34+ HSCs (BLT)	Robust and long-term engraftment of human NK cells with increased expression of IFN-γ, TNF-α, perforin, Ki67, and CD107a after HIV infection in vivo. Less susceptible to GvHD. May be a suitable long-term model to study the effects of NK cells on the HIV reservoir.
NOG-EXL	GM-CSF IL-3	HuCD34 [113,114]	Human CD34+ HSC	Robust myeloid cell engraftment, but low NK cell reconstitution. Acute HIV infection decreased levels of CD56^bright^ NK cells.
NSG-SMG3	GM-CSF IL-3 SCF	HuPBL HuCD34 [115] BLT [116,117]	Human PBL (HuPBL), CD34+ HSC (HuCD34), or human thymus, liver and CD34+ HSCs (BLT)	Robust myeloid cell engraftment, and thereby NK cell development. Early death due to GvHD-related macrophage activation syndrome.
MISTRG	GM-CSF IL-3 M-CSF TPO	HuCD34 [80,118]	Human CD34+ HSC	Robust engraftment of myeloid cells, which aid in engraftment of human NK cells. Robust T cell engraftment with efficiently infection by X4 and R5 tropic HIV isolates. Prone to GvHD-related severe anemia 20 weeks post-transplantation.
SRG-15	IL-15	HuCD34 [49,110]	Human CD34+ HSC	Significantly improved engraftment of human NK cells compared to HuCD34-SRG or HuCD34-MISTRG mice. Appropriate long-term pre-clinical model to test curative therapies against the HIV reservoir.
MISTRG-	GM-CSF IL-3 IL-6 IL-10 M-CSF TPO	HuCD34 [97]	Human CD34+ HSC	Significantly improved myeloid and NK cell engraftment and function compared to HuCD34-NSG mice. Human NK cells during acute infection, ART suppression, and rebound infection show functional profile patterns similar to clinical studies. Importantly, human NK cells were shown to directly control HIV-1 replication in vivo.

**Table 3 microorganisms-11-01984-t003:** Other recent humanized mouse models to study NK cells.

Mouse Strain	Human Cytokines Expressed	Mouse Model	Xenografts	Summary of Key Findings
NOG-IL15	IL-15	PBL [119]	Human PBL	Robust engraftment of human NK cells with subsequent exhaustion and dysfunction, likely due to supraphysiologic level of human IL-15. Prone to GvHD. In vivo HIV infection has not been studied yet.
hIL-7xhIL-15 KI NSG	IL-7 IL-15	HuCD34 [124]	Human CD34+ HSC	Abnormally high frequencies of human NK cells, likely due to supraphysiologic level of human IL-15. In vivo HIV infection has not been studied yet. Likely prone to GvHD.
NSG-SMG3-IL15	GM-CSF IL-3 IL-15 SCF	None		Transgenic expression of IL-15 in NSG-SGM3 mice. Further studies to assess NK cells and HIV infection in vivo have yet to be performed.

## Data Availability

Not applicable.

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
