# Peer review of "Current Advances in Humanized Mouse Models for Studying NK Cells and HIV Infection"

_microorganisms, 2023, doi:10.3390/microorganisms11081984_

Round 1

Reviewer 1 Report

The authors have assembled an impressive collection of previous attempts at improving myeloid reconstitution in humanized mice.  In general, the paper is very well written with clear language and comprehensive statements.  One aspect I would point out however is that the latent reservoir may consist of cells beyond CD4+ T cells (including macrophage).  The point made about the slow loss of the reservoir over time is independent of proliferation of latently infected cells based on a reference from 1999 but more recent work looking at integration sites would suggest that substantial persistence is caused by proliferation of infected cells.

Reviewer 2 Report

This review article describes humanized models to study HIV infection. Here are my comments:

-Authors need to provide a cartoon showing how HIV infection affects different immune cells which are relevant to this review article. e.g. NK cells, CD4+ T cells, etc

-Authors also need to provide a table showing the use of different antiretroviral therapies in the humanized mice models relevant to HIV infection and their failure/efficacy.

-Authors mention in their review that memory CD4+T cells can stay in the lymphoid organs for years in a quiescent state under antiretroviral therapy and get reactivated when therapy is stopped; how can this phenomenon be studied in humanized models relevant to HIV infection. Will humanized mice survive long to study this phenomenon. Please incorporate this in discussion.

Minor errors

Reviewer 3 Report

 HIV infection is a major global health challenge and has resulted in millions of people being infected. In the current manuscript, Jocelyn T. Kim and the colleagues reviewed and summarized the advances and limitations in humanized mouse models of HIV study, especially in NK cells.  

This is a very good review article, it meets the publication criteria, and recommend to publish in current form. 

Reviewer 4 Report

This review article by Kim, Bresson-Tan, and Zack is an excellent summary of the currently available humanized mouse models for HIV research.  The manuscript includes models for studying NK cells, which are important for HIV control.  The manuscript is well-written, concise, and logically organized.  Investigators will easily find the information they need about humanized mouse models.  There are some minor corrections listed below.  The authors may consider changing how the information in the tables is presented.  As shown, the table entries are centered lines, which makes it harder to read the longer strings of text.  Consider left-justifying each entry.  Furthermore, the focus of this manuscript is the mouse strains used for various human cell/tissue grafts.  A large spreadsheet that includes the short name of each strain (i.e. NOG, NSG-BLT, SRG, MISTRG, etc.), their mouse strain designation, the human cytokines and chemokines expressed, their xenografts, and other features, would be very useful.  This spreadsheet could be a supplemental figure.      

Line 40: is there a citation about organoid platforms for HIV to include here?  A review article would be helpful.

Line 48, remove the paragraph break.

Line 95, remove the comma after “transplantation”.  Add an “s” to “T cell” at the end of the line.

Lines 164-176: this paragraph does not stand on its own.  Either remove the paragraph break or change the topic sentence.

Line 178, specify that the recent changes in abortion and fetal research policies apply to the U.S.

Line 182, change “is” to “are” or remove the “s” from “disadvantages”.

Line 203, the heading for section 3 (Improved mouse models for HIV-1 and NK research) appears to be out of place.  Move it to Line 190.  The paragraph on lines 190-202 is not about NeoThy mice, so it belongs in Section 3.

Line 214, remove the paragraph break.

Line 223, insert “mice” after “NSG-BLT”.

Line 224, change “NK cells demonstrated inhibition of viral replication” (too wordy) to “NK cells inhibited viral replication”.

Line 253, remove the word “can”.

Line 283, remove the phrase “In addition”.

Line 323, correct the typo in “mimemtic”.

Line 345, insert “they have not been” to replace “not”.

Table 3, insert “be” in the third Summary entry in the phrase “HIV infection in vivo have yet to performed”.

Line 361, change “with” to “to”.

Line 371, add an “s” to “vaccine”.

The quality of English is very good.  A few minor grammar edits and typo corrections are needed.

Round 2

Reviewer 2 Report

No further comments, concerns addressed properly